# A 1-to-3 GHz 5-to-512 Multiplier Adaptive Fast-Locking Self-Biased PLL in 28 nm CMOS

**Binghui Wang** [1,2], **Haigang Yang** [3,*] **and Yiping Jia** [4]

1 Aerospace Information Research Institute, Chinese Academy of Sciences, Beijing 100190, China; wangbinghui18@mails.ucas.ac.cn
2 School of Electronic, Electrical and Communication Engineering, University of Chinese Academy of Sciences, Beijing 100094, China
3 School of Microelectronics, University of Chinese Academy of Sciences, Beijing 100049, China
4 Shandong Industrial Institute of Integrated Circuits Technology Ltd., Jinan 250001, China; jiayp@cwisemicro.com
* Correspondence: yanghg@mail.ie.ac.cn

**Abstract:** Based on a self-biased architecture, this paper presents a novel adaptive fast-locking, wide operating range and low-jitter phase-locked loop (PLL). A current injection and adaptive bandwidth technology with minimum area overhead is employed to speed up the loop equilibrium acquisition process, without any adverse impact on the steady-state loop dynamics and the jitter performance. The proposed start-up circuit resets the loop to an appropriate initial state in order to shorten the initial ramp-up interval of the voltage-controlled oscillator (VCO), also resulting in cutting down the pull-in time. In addition, a proportional factor is introduced to give some kind of flexibility in the circuit design optimization. The proposed adaptive fast-locking self-biased PLL (AFL-SPLL) is designed and realized in a prototype based on TSMC 28 nm CMOS process, having a supply voltage of 0.9 V and an area of 0.0281 mm$^2$. This PLL demonstrates a tuning range of 1 to 3 GHz and power consumptions from 0.91 mW at 1 GHz to 4.6 mW at 3 GHz operating frequency. The experimental results show that the capture process has been accelerated by up to 84.7% over large division ratios, yet the capture performance did not deteriorate at all for small division ratios. Meanwhile, the circuit implementation gave almost no area increase and yet achieved a reduction in the lock-in time of about 6.5 times, namely from 23.5 μs (without the adaptive locking) to only 3.6 μs (with the adaptive locking) on the maximum operation frequency condition of 3 GHz.

**Keywords:** self-biased PLL; adaptive fast-locking PLL; start-up circuit; current injection





## 1. Introduction

The PLLs have been widely utilized in high-speed data transmission systems such as SerDes, wireless transceivers and disk read/write channels, where short-phase locking time and low-phase noise are required.

For conventional PLLs, the division ratio of feedback divider (FD) affects the bandwidth and the damping factor ($\zeta$), which, for some of the settings, may lead to poor jitter performance or even unstable states. A typical PLL tends to lock on the target state at a roughly constant speed due to the fixed bandwidth. The self-biased PLLs, which find the optimal operating bias level and charge pump (CP) current in an adaptive fashion [1,2], feature high stability, low jitter, and large operation range, and have been widely used [3–6]. Further, the self-biased PLLs are in no need of resistors and are independent of operating process, voltage, temperature (PVT) and frequency.

In order to realize the fast-settling process of the PLLs, two CP currents of different magnitudes were adopted [6,7], where the larger CP current was utilized under the acquisition process, and the smaller one was utilized in steady state. Therefore, a shorter pull-in time can be achieved due to the wider bandwidth in the capture process. However, the

CP currents were fixed, leaving the pull-in performance to fluctuate over a wide range of division ratios.

To address such an issue, the concept of adaptive charge pump current was put forward to allow the bandwidth being adjusted adaptively [8,9]. In [8], the charge/discharge current of the CP varies appropriately over the CP output voltage range, which can effectively speed up the capture process by 72% in comparison. However, such a technique cannot be applied to a self-biased architecture where the CP current is adaptive. The adaptive current of the CP in [9] is determined by the output result from the phase frequency detector (PFD), which can shorten the lock-in time from 8 us to 2 us. Unfortunately, the resistors used in such a design would otherwise make the self-biased PLL's performance deteriorate with the process and temperature variations.

There are some other effective fast-locking architectures, as proposed in [10–14]. The sub-sampling PLL [10] is a dual-loop architecture with an FLL, which speeds up the locking time and saves half of the power budget during the frequency acquisition. This fast-locking technology is more efficient with those PLLs of employing the push–pull sub-sampling phase detector (SS-PD) or the bang-bang phase detector but not PFD, so it will offer limited benefits on the fast-locking performance of the self-biased PLLs. The proposed PLL in [11] utilizes an aperture phase detection (APD) mechanism and a dead zone creator to save the power consumption in locked state and reduce the locking time by 32%, where the acceleration is relatively weak. Some other phase error compensation and bandwidth control techniques proposed in [12–14] help to significantly reduce the locking time, but their relative circuits will greatly increase the logic complexation and area.

As just mentioned, it is a great challenge to optimize the self-biased PLL design for a shorter locking time and overhead area. In this paper, an adaptive fast-locking self-biased PLL with a start-up circuit and an auxiliary adaptive fast-locking current circuit (AFLCC) is proposed, with loop dynamics parameters that are characteristically independent of the operating frequency and the division ratio. A scaling factor is also introduced to facilitate circuit design optimization. As a result, the AFLCC speeds up the loop equilibrium acquisition process with minimum area overhead and power consumption, meanwhile avoiding any adverse impact on the steady-state loop dynamics and the jitter performance.

## 2. Self-Biased PLL Fundamentals

The conventional PLL mainly consists of a phase-frequency detector (PFD), a charge-pump (CP), a loop filter (LF), a voltage-controlled oscillator (VCO), and a feedback divider (FD) (see Figure 1). The key dynamics parameters of the conventional second-order PLL are defined as

$$\omega_N = \sqrt{\frac{I_{CP}K_{VCO}}{C_1 N}}, \; \zeta = \frac{1}{2}\sqrt{\frac{I_{CP}K_{VCO}R^2 C_1}{N}} \tag{1}$$

where $R$ and $C_1$ are the LF resistor and capacitor, respectively, $I_{CP}$ is the CP current, $K_{VCO}$ is the VCO gain, $N$ is the FD ratio, $\omega_N$ is the bandwidth, and $\zeta$ is the damping factor.

For typical PLLs, the parameters $I_{CP}$, $R$, $C_1$, $N$, and $K_{VCO}$ are usually fixed, leading to fixed $\omega_N$ and $\zeta$. Moreover, $\omega_N$ represents the response rate of the system. Consequently, the typical PLLs cannot always obtain an optimally fast acquisition under all the operating conditions.

The yellow box in Figure 1 shows the diagram of the basic self-biased PLLs, which mitigates the drawbacks (fixed $\omega_N$ and $\zeta$) of the typical PLLs [1]. $I_{CP}$ is set to some multiplied $x$ of the VCO buffer tail current $2I_D$, and the resistor in LF is realized by a $1/g_m$ resistance, which is proportional to the output period, such that constant $\zeta$ and $\omega_N/\omega_{ref}$ can be attained and simply given by Equations (2) and (3):

$$\frac{\omega_N}{\omega_{ref}} = \frac{\sqrt{\frac{I_{CP}K_{VCO}}{C_1 N}}}{\frac{f_{VCO}}{N}} = \frac{\sqrt{xN}}{2\pi}\sqrt{\frac{C_B}{C_1}} \tag{2}$$

$$\zeta = \frac{1}{2}\sqrt{\frac{I_{CP}K_{VCO}R^2C_1}{N}} = \frac{y}{4}\sqrt{\frac{x}{N}}\sqrt{\frac{C_1}{C_B}} \tag{3}$$

where $y$ is the ratio of the $1/g_m$ resistance to the symmetric loads used in the VCO buffer stages, $C_B$ represents the equivalent capacitance of the VCO, and $f_{VCO}$ is the operating frequency of the VCO.

In [2], an inverse-linear current mirror circuit is adopted to make x equal to $1/N$, and a sampled filter circuit is used to make y equal to $Q_O/Q_I$ for the feedforward network, where $Q_O$ and $Q_I$ is the output charge and input charge of the feedforward network respectively. As a result, $\omega_N/\omega_{ref}$ and $\zeta$ are updated and given by

$$I_{CP} = \frac{1}{N}(2I_D) \tag{4}$$

$$\frac{\omega_N}{\omega_{ref}} = \frac{1}{2\pi}\sqrt{\frac{C_B}{C_1+C_2}}, \; \zeta = \frac{1}{4}\sqrt{\frac{C_B(C_1+C_2)}{C_2}} \tag{5}$$

Therefore, the self-biased PLLs have constant loop dynamic parameters, $\omega_N/\omega_{ref}$ and $\zeta$, which are independent of $N$, operating frequency, and PVT.

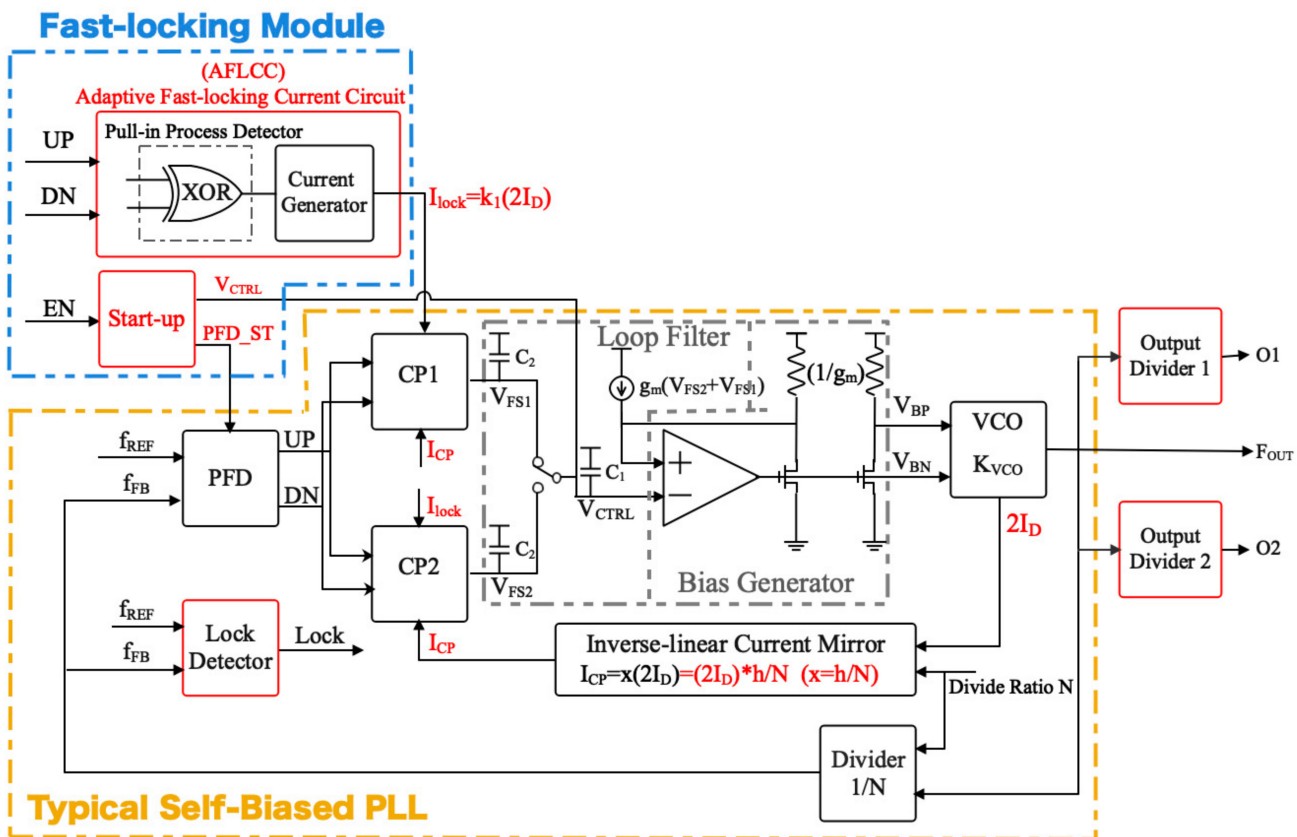

**Figure 1.** The proposed AFL-SPLL block diagram.

## 3. Proposed Adaptive Fast-Locking Architecture for the Self-Biased PLL

Self-biased PLLs show good performance in stability, process independence, jitter, and wide frequency range [1,2]. However, the inverse-linear current mirror circuits possess the property where the larger $N$ is, the smaller $I_{CP}$ is. In other words, if $N$ is extremely large, the pull-in process would be extremely slow due to the extremely narrow bandwidth. Therefore, the lock-in time can vary with different operating configurations.

Figure 1 suggests a solution to circumvent this problem. A fast-locking module based on the current injection and adaptive bandwidth technology is adopted. When the loop is in the pull-in process, $I_{CP}$ will be increased adaptively in a way to widen the bandwidth, and when in the locked state, $I_{CP}$ is kept at an optimized value to acquire low jitter and high stability. Further, the lock detector and output dividers are included to work in conjunction with other modules in AFL-SPLL. The output dividers 1 and 2 perform a division of 1–30, with both O1 and O2 operating over a frequency range from 333.3 MHz to 3 GHz. This way, the output has a wide frequency range, yielding more choices for the system speed grade.

The start-up circuit serves the purpose of discharging and presetting the voltage $V_{CTRL}$ to a proper level quickly, which facilitates activating VCO to oscillate reliably. Specifically, the complete loop will not finish the set-up phase until the start-up circuit generates a flag signal indicating that the VCO is now being steadily oscillated.

The AFLCC will pump an extra adaptive current $I_{lock}$ into CP, which widens the bandwidth and accelerates the pull-in process. Like the $I_{CP}$, $I_{lock}$ is also adaptable and given by

$$I_{lock} = k_1(2I_D) = \frac{k_1 N}{h} I_{CP} \tag{6}$$

To allow some flexibility in circuit design optimization, the parameter x is scaled to $(h/N)$ instead of $(1/N)$. Assuming the original dynamics in the steady-state to be $\zeta_0$ and $\left(\omega_N / \omega_{ref}\right)_0$, both parameters in the capture process should be updated such that

$$I_{CP} = \frac{h}{N}(2I_D) \tag{7}$$

$$\zeta = \frac{1}{2}\sqrt{\frac{(I_{CP} + I_{lock})K_{VCO}R^2(C_1 + C_2)}{N}} = \frac{\sqrt{h + k_1 N}}{4}\sqrt{\frac{C_B(C_1 + C_2)}{C_2}} = \sqrt{h + k_1 N}\zeta_0 \tag{8}$$

$$\frac{\omega_N}{\omega_{ref}} = \sqrt{\frac{2(x + k_1)I_D K_{VCO}}{(C_1 + C_2)N}} = \frac{\sqrt{h + k_1 N}}{2\pi}\sqrt{\frac{C_B}{(C_1 + C_2)}} = \sqrt{h + k_1 N}\left(\frac{\omega_N}{\omega_{ref}}\right)_0 \tag{9}$$

Both parameters are multiplied by $\sqrt{h + k_1 N}$. Thus, compared with the original self-biased architecture for a certain operating frequency, $\omega_N$ was in effect enlarged by $\sqrt{h + k_1 N}$, furthering to speed up the acquisition. The AFLCC is only activated in the pull-in process but should be shut in the locked state.

In other words, the proposed fast-locking module has no implication whatsoever on the locked state behavior of the PLLs.

### 3.1. Adaptive Fast-Locking Current Circuit (AFLCC)

The proposed AFLCC, shown in Figure 2, adopts the current mirror structure. The transistors M1–M4 form a replica of the bias generator circuit to keep $I_{lock}$ following $I_D$ precisely. The operating principle is shown in Figure 3. The XOR gate can detect the state of the loop through the outputs of the phase-frequency detector (PFD). When the PLL is in the capture process, the XOR generates a pulse (SP_EN) and produces a locking current. When in the locked state, M3 and M7 can cut off the locking current.

Obviously, the switch transistors are not ideal and induce a small delay $t_{delay}$ at the output $I_{lock}$. Intuitively, every time the AFLCC is turned on, it will take some time to pull the voltage $V_{LCPNBS}$ from 0 to a correct value. So, no current will be generated until $V_{LCPNBS}$ rises above the transistor threshold voltage, as shown in Figure 3.

Actually, this intrinsic small delay is desired. The PFD always outputs a pulse even in the locked state. Therefore, if the transistors ideally induce no delay, AFLCC would always tend to generate an output current, large or small, so that the multiplied dynamics parameters may cause the stability problem.

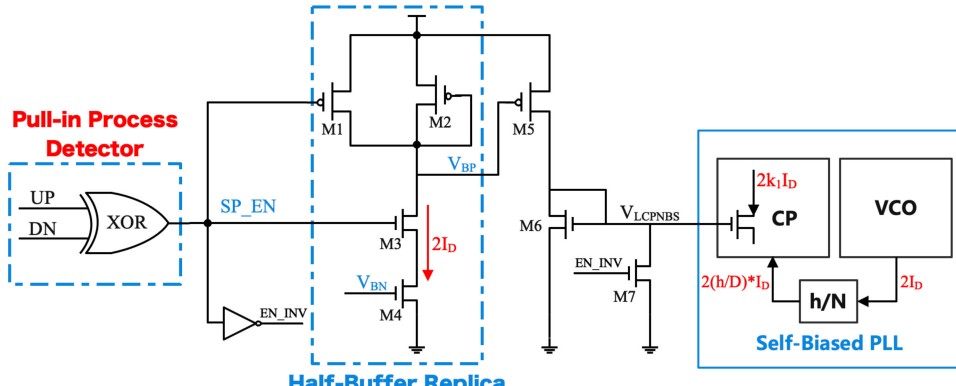

**Figure 2.** The fast-locking current circuit.

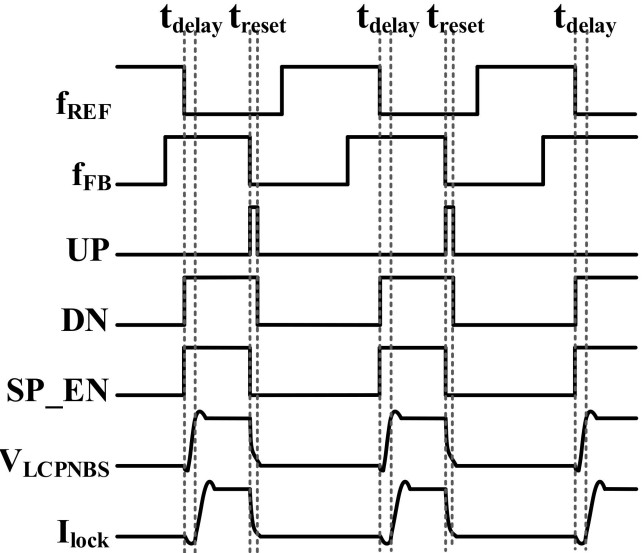

**Figure 3.** The principle of the adaptive fast-locking circuit.

The AFLCC can be disabled earlier before the real lock-in starts due to the intrinsic delay of switches. If the phase error becomes smaller than a certain value or minimum pulse width [14], the AFLCC will not be activated at all. This working principle will benefit the pull-in process, particularly for the operating conditions of high reference frequency or of the loop getting close to the locked state.

Under the high reference frequency condition, $N$ is small so that $I_{CP}$ holds a large value by (7). Therefore, the bandwidth will be wide enough, and the loop could exhibit fast pull-in performance with no need of extra current injection. Additionally, the phase error in terms of time is always small due to the small reference period, in which case the intrinsic delay induced by the switches makes AFLCC cease having any influence on the loop. This way, no degradation of the original good lock-in performance will ever happen.

When the loop comes close to the locked state, the phase error also decreases. If an extra large $I_{lock}$ is added to $I_{CP}$ at the time, the loop would be likely to repeatedly ripple around the steady point. This is because the extra $I_{lock}$ may make the frequency step exceed the lock-in range $\omega_L$, taking the PLL out of the lockable region. The $\omega_L$ is equivalent to the maximum frequency error for which acquisition is almost instantaneous. Therefore, the delay is helpful in damping the margin ripple in the pull-in process.

In other words, the AFLCC can widen the bandwidth adaptively following the operating frequency and $N$, giving rise to a much shorter lock-in time of the system, especially for larger $N$.

### 3.2. The Start-Up Circuit

The start-up circuit, shown in Figure 4, consists of logic gates and a counter, which can start the VCO steadily and guarantee the initial frequency error is well within the pull-in range $\omega_P$, in a way to enhance the robustness of the system.

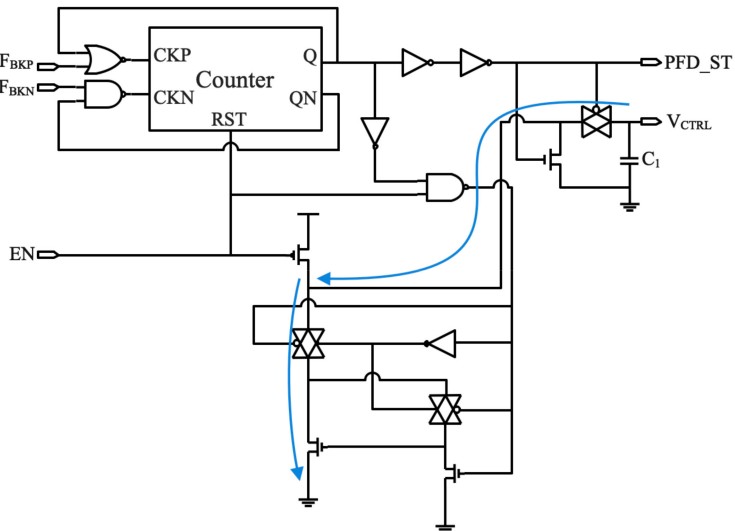

**Figure 4.** The diagram of the start-up circuit.

The counter decides whether the VCO has reached a normal oscillation, and when satisfied, asserts the flag signal PFD_ST. The period of the counter can be set appropriately—typically, 64 cycles in this design.

The working principle and simulation waveform are shown in Figure 5a. The start-up circuit initially sets $V_{CTRL}$ to the supply voltage, and the enable signal EN is set low. If *EN* is asserted to a high level, $C_1$ begins to discharge, and $V_{CTRL}$ drops slowly. The path of discharge current is marked with the blue arrow in Figure 4. When $V_{CTRL}$ falls to a proper value, the VCO starts oscillating, and the counter will be activated. Until the VCO oscillates continuously over 64 periods, PFD_ST will be pulled to the high level. At this point, while the start-up circuit is disabled, the complete loop is set up, and the PLL comes into the pull-in process.

The different counter value ($N_C$) can incur different lock-in times. As VCO tends to start oscillating at a fixed voltage, it should be appropriate to set up the whole loop at a lower voltage of $V_{CTRL}$, where the operating point is situated in the linear region of the $K_V$ curve and the loop bandwidth is larger. Thus, the pull-in process can effectively be accelerated. Figure 5 shows the simulation waveform where $F_{ref}$ is set to 300 MHz and $N$ is set to 10 equally, and $N_C$ is set to 64 and 16, respectively. The result indicates that the former corresponds to a faster pull-in performance.

Notably, although speeding up the discharge on the capacitor $C_1$, or prolonging the discharge time, can accelerate the capture process, it is important to avoid $V_{CTRL}$ from dropping too rapidly and hence going out of the tuning range during the counting process. Preferably, the optimal point to set up the whole loop would be at the center frequency $f_{op}$ of the VCO.

Assuming the VCO starts at the moment $t = t_0$ when $V_{CTRL} = V_0$, $f_{VCO} = f_0$, and the expected time to set up the loop is the middle point within the VCO tuning range where $V_{CTRL} = V_{op}$, $t = t_{op}$, and $f_{VCO} = f_{op}$. Additionally, assuming the discharge process of $C_1$ is approximately linear from $t_0$ to $t_{op}$, and the equivalent frequency is $(f_0 + f_{op})/2$, we can formulate Equations (10) and (11) and calculate the optimal $N_C$ by Equation (12).

$$V_0 = VDDe^{-\frac{t_0}{RC}}, \quad V_{op} = VDDe^{-\frac{t_{op}}{RC}} \tag{10}$$

$$t_0 - t_{op} = \frac{2}{f_0 + f_{op}} * N_C \tag{11}$$

$$N_C = \frac{f_0 + f_{op}}{2} * VDD * \left(e^{-\frac{t_0}{RC}} - e^{-\frac{t_{op}}{RC}}\right) \tag{12}$$

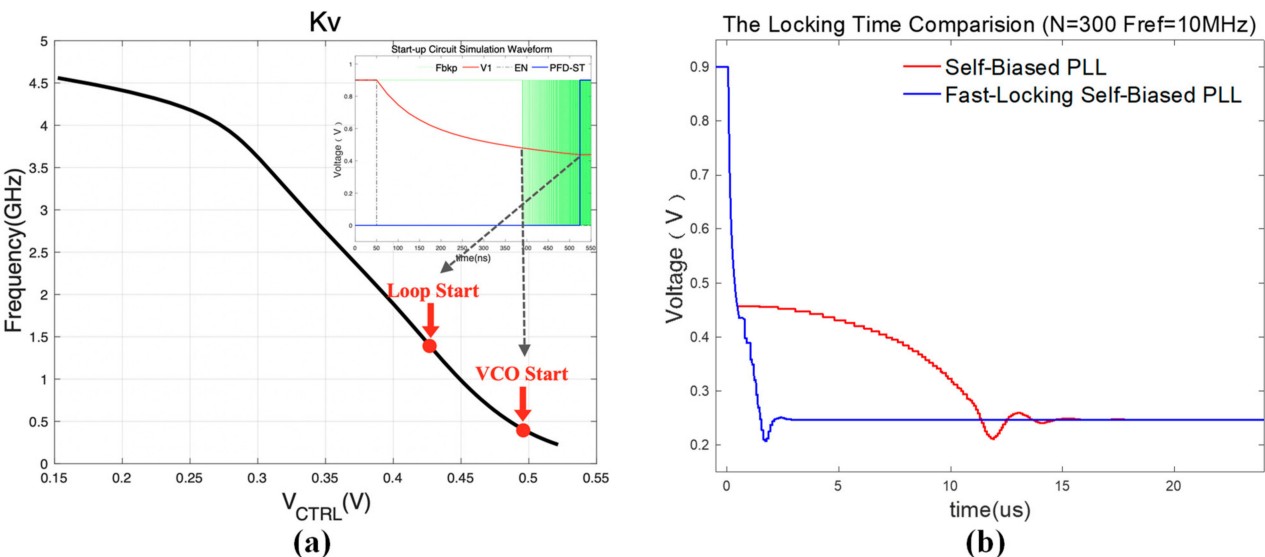

**(a)**

**(b)**

**Figure 5.** (**a**) The $K_V$ curve and simulation waveform. (**b**) Locking time of fast-locking self-biased PLL with different counter values (16 and 64).

## 4. Experimental Results

### 4.1. Layout

The AFL-SPLL has been designed and implemented in a standard 28 nm CMOS process with a supply voltage of 0.9 V. Figure 6 shows the layout with the fast-locking module included, which is marked with yellow box. The total core area of the fast-locking PLL is merely 223 μm × 126 μm (=0.0281 mm$^2$), in which the fast-locking module is measured as 260 μm$^2$, about 0.93% of the total PLL area. Apparently, such area overhead is negligible.

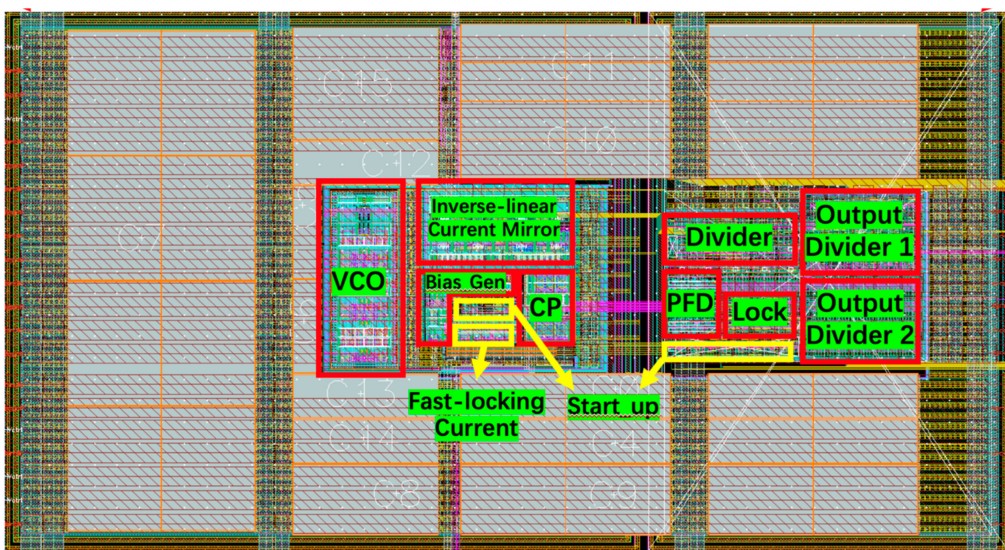

**Figure 6.** Layout of the fast-locking PLL.

### 4.2. Pre-Layout Simluation Result

The proposed fast-locking self-biased PLL utilizes AFLCC and the start-up circuits to achieve improved performance in the loop acquisition. To verify the loop stability, accelerated lock-in performance, etc., simulations were carried out for operating frequencies at 1 GHz/3 GHz and with varying *N*. The VCO module holds the linear frequency range from 1 to 3 GHz. In addition, two key loop dynamics parameters, $\zeta$ and $\omega_N/\omega_{ref}$, were calculated by Equation (1) with the simulation data.

The simulation results are compared in Figure 6, which reveals that the proposed fast-locking module can shorten the lock-in time of the self-biased architecture evidently and push them to a similar order of magnitude as a result of the adaptive bandwidth and the proper counter value ($N_C$) in the start-up circuit.

The transient result in Figure 7a,c also confirms that the pull-in time of the original self-biased PLLs would become excessively long in the cases where *N* is large. Additionally, the lock-in time is approximately proportional to *N*. Equation (13) in [1] further reveals that the lock-in time is inversely proportional to $I_D$. Considering that $I_D$ is inversely proportional to the division ratio *N*, the pull-in time tends to be proportional to *N*. However, this equation neglects the saturation behavior in sampling, where the sampling capacitor may be fully charged/discharged with its voltage reaching to the supply/ground. By exploiting such saturation behavior, the locking time can be reduced to its maximum effect.

$$t = |V_{CTRL}(t) - V_{CTRL}(0)| * \frac{2C_1}{I_D} \tag{13}$$

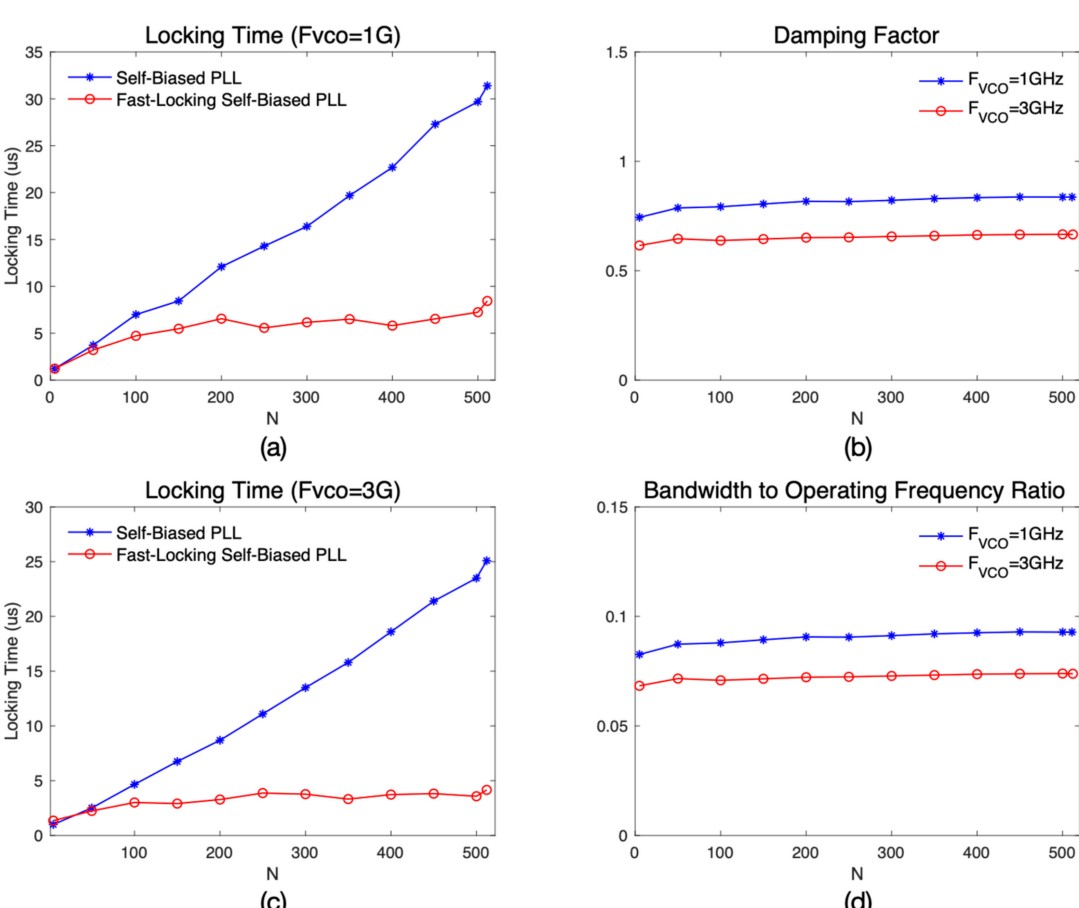

**Figure 7.** (**a**) Lock-in time under 1 GHz operating frequency, (**b**) $\zeta$ under 1 and 3 GHz operating frequency, (**c**) lock-in time under 3 GHz operating frequency, and (**d**) $\omega_N/\omega_{ref}$ under 1 and 3 GHz operating frequency.

Figure 7b,d also reveal some characteristics of the self-biased architecture. As shown, the loop dynamics parameters $\zeta$ and $\omega_N/\omega_{ref}$ remain roughly constant for all the division ratios (5–512), which are approximately 0.7 and 0.008, respectively. Those small deviations at the same N are mainly caused by inaccuracies in the inverse-linear current mirror.

### 4.3. Post-Layout Simulation Result

The current consumptions of each module were measured in our post-layout simulation. Then, the power dissipation was estimated (see Figure 8). The proposed AFL-SPLL has a tuning range of 1 to 3 GHz and typical power consumptions of 0.91 mW at a 1 GHz operating frequency, or 4.6 mW at a 3 GHz operating frequency. As estimated in Figure 8, the VCO module dissipates more than half of the total power, i.e., 58.72%. In fact, the extra fast-locking module neither contributes to much of the area overhead (0.93%), nor consumes much of the power (0.28%).

The post-layout simulation results relating to the reference spur and phase noise are shown in Figure 9, where $F_{ref}$ and $N$ are set to 200 MHz and 10, respectively. The reference spur and the phase noise are leveled, namely $-50.63$ dBm versus $-51.06$ dBm, $-110$ dBc/Hz versus $-113$ dBc/Hz, on conditions with and without the fast-locking module. Because of the similar loop dynamics at the locked state, the AFL-SPLL is on a par with a typical self-biased PLL in terms of jitter and phase-noise performance.

The post-layout simulation was executed under typical and four extreme conditions (different corner and temperature), and its transient result is shown in Figure 10. Clearly, the lock-in time can be significantly shortened to about 80% (at the ss corner).

Table 1 shows the comparison results regarding the performance characterization. As seen, the proposed PLL not only retains the advantages of the self-biased architecture including large operating range, low jitter, etc., but also improves the capture performance significantly at expenses of minimum area overhead. Compared with those of previously reported fast-locking PLLs, the VCO frequency range in the proposed PLL is typically larger compared to the other works, with relatively less area occupied. Importantly, the locking time of this work has been reduced to 1.23–8.45 us (shown in Table 1).

**Table 1.** Performance comparison.

| REF | [10] | [11] | [14] | [15] | This Work |
|---|---|---|---|---|---|
| Process (nm) | 65 | 180 | 65 | 45 | 28 |
| Supply (V) | 1.0 | 1.8 | 1.2/1.5 | 0.9 | 0.9 |
| Structure | SS-PLL | Dual loop | CPPLL | SIL-TPLL | Self-biased |
| Ref Freq (MHz) | 103 | 50 | 1–800 | 150 | 1.95–600 |
| Division Ratio | 32 | 64 | - | 16 | 5–512 |
| VCO Freq (GHz) | 3.296 | 3.2 | 0.2–1.6 | 2.4 | 1–3 |
| PN@1 MHz (dBc/Hz) | $-128.4$ | $-118$ (Out-of-band) | - | - | $-113.0$ |
| Ref Spur (dBc/Hz) | $-82.2$ | $-74$ | $-54$@800 M | $-40.4$ | $-51.06$ |
| Lock-in time (us) | <10 | 1.9 | <3 | 1.8 | 1.23–8.45 |
| Area (mm$^2$) | 0.21 | – | 0.05 | 0.013 | 0.0281 |
| Power (mW) | 7.53 | 4.15 | – | 5.6 | 0.91 @1 GHz 4.60 @3 GHz |
| Meas/Sim | Mea | Sim | Sim | Mea | Sim |

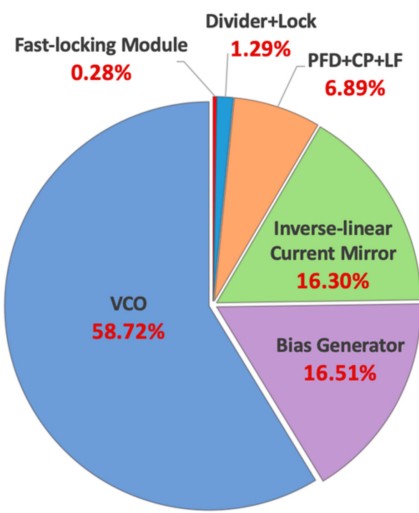

**Figure 8.** Power consumption of the proposed AFL-SPLL.

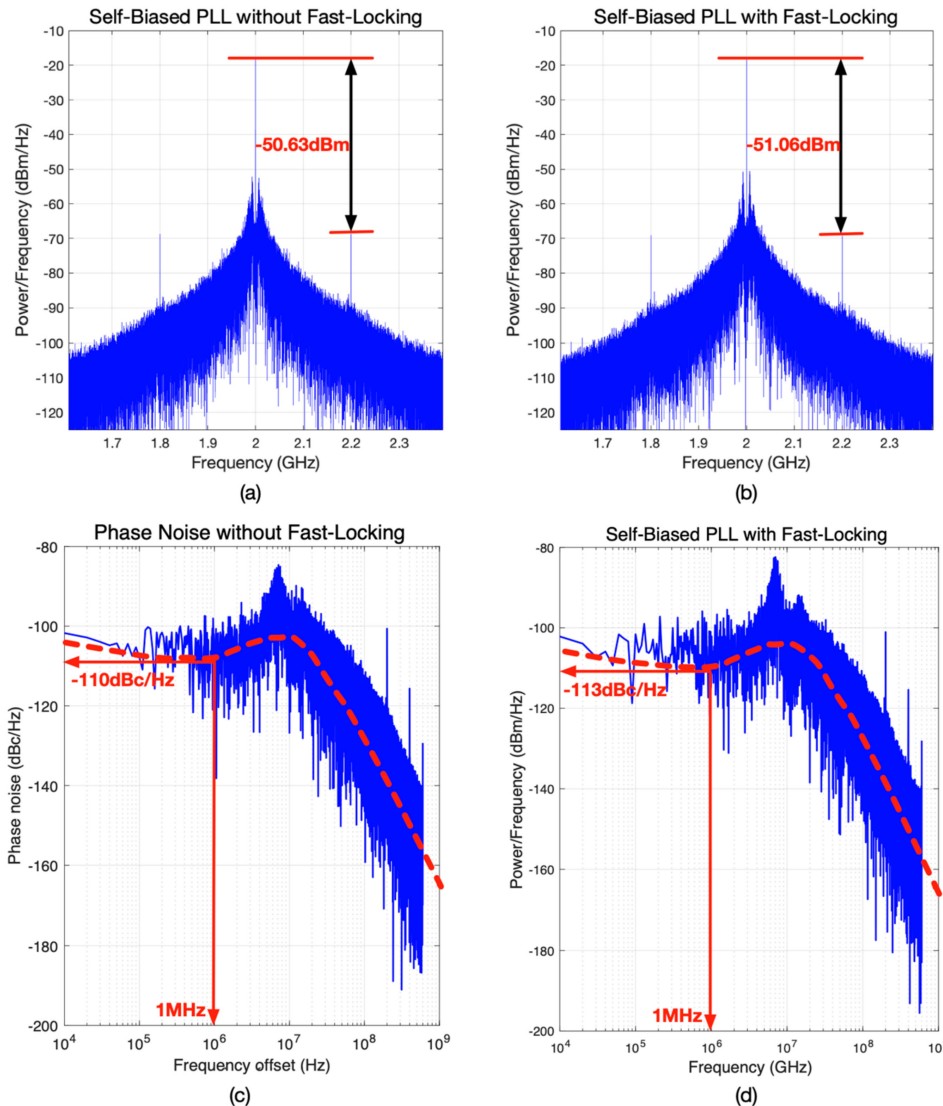

**Figure 9.** (**a**) Simulated reference spur of the PLL without fast-locking module, (**b**) Simulated reference spur of the PLL with fast-locking module, (**c**) l Simulated phase noise of the PLL without fast-locking module, and (**d**) Simulated phase noise of the PLL with fast-locking module.

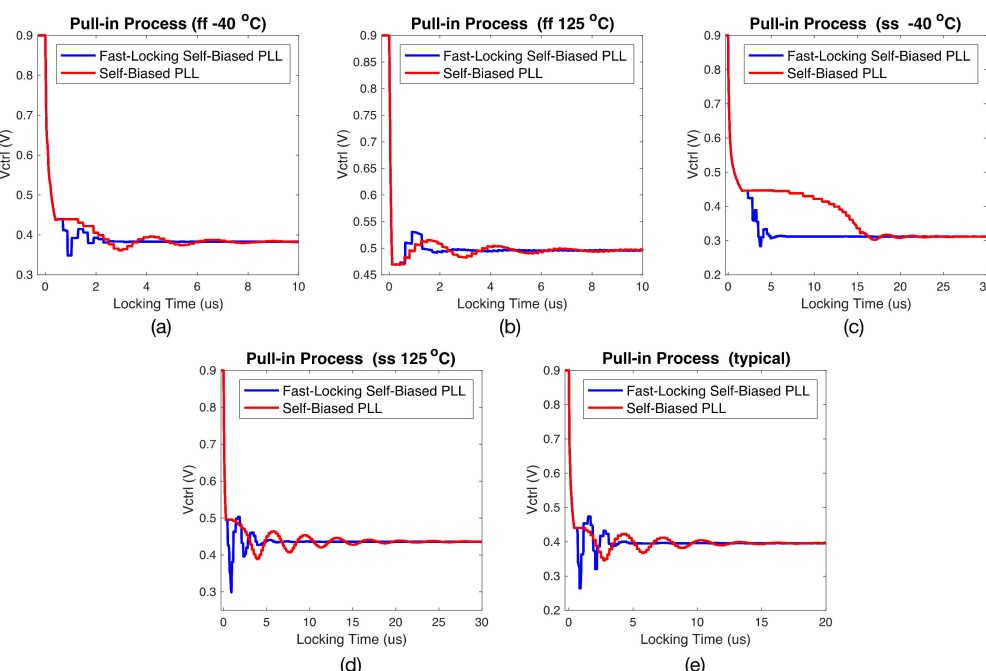

**Figure 10.** (**a**) Pull-in process of the self-biased PLL with and without fast-locking module in ff corner and −40 °C, (**b**) Pull-in process of the self-biased PLL with and without fast-locking module in ff corner and 125 °C, (**c**) Pull-in process of the self-biased PLL with and without fast-locking module in ss corner and −40 °C, (**d**) Pull-in process of the self-biased PLL with and without fast-locking module in ss corner and 125 °C, and (**e**) Pull-in process of the self-biased PLL with and without fast-locking module in typical condition (f$_{vco}$ = 2 GHz, N = 500).

## 5. Conclusions

A fast-locking system for the self-biased PLLs, consisting of a start-up circuit and an AFLCC, is designed and layout is implemented in 28 nm COMS technology with 0.9 V supply, having a tuning range from 1 GHz to 3 GHz and working at a division ratio anywhere between 5 to 512. The core area of the fast-locking PLL is just 0.0281 mm$^2$ with only 0.93% of the area occupied by the fast-locking module. The lock-in time is shortened by about 84.7% for large division ratios without sacrificing jitter performance. In addition, a proportional factor h and the optimal counter value $N_C$ of the start-up circuit are introduced to provide extra flexibility in optimizing the PLL design to some required specifications. As demonstrated in the post-layout simulation results, the proposed fast-locking PLL has achieved faster lock-in times on the various loop dynamics conditions, while the loop stability is well-maintained.

It is noted that the capture duration of a typical self-biased PLL is largely affected by the PVT conditions. The proposed AFL-SPLL alleviates this dependency to a certain extent but cannot completely eliminate such an impact of the PVT variations on its locking time. This circuit design issue requires further investigation.

**Author Contributions:** H.Y. supported and conceived the experiments and correspondence to this article; B.W. performed the numerical simulations and the experiments, as well as most of the analysis, data acquisition and processing, and writing; Y.J. contributed to parts of the analysis and discussion. All authors have read and agreed to the published version of the manuscript.

**Funding:** This work was supported by the National Natural Science Foundation of China under Grant 61876172.

**Conflicts of Interest:** The authors declare no conflict of interest.

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
