# Peer review of "A 1-to-3 GHz 5-to-512 Multiplier Adaptive Fast-Locking Self-Biased PLL in 28 nm CMOS"

_electronics, doi:10.3390/electronics11131954_

Round 1

Author Response

Thanks, we very much appreciate the reviewer’s supporting comments.

Reviewer 2 Report

An adaptive fast-locking self-biased phase locked loop is designed using 28 nm CMOS process. The tuning range is from 1-3 GHz, which makes it suitable for microwave applications. The novelty lies in suggesting an adaptive fast-locking current circuit to make the PLL parameters achieve independency from the operating frequency and frequency division ratio. Comparative studies with the recent literature clearly demonstrate the benefits of the proposed circuit topology. Authors may consider the following suggestions to further improve the work.

1. It is claimed in the Abstract that the PLL has been fabricated. However, the paper does not provide supporting evidence and results of the actual fabricated PLL; rather, only the simulation results are shown. Authors are encouraged to include the photograph of the fabricated chip, and show detailed comparisons of the experimental measurement results with those of the simulated ones.

2. The Introduction is too short and abruptly ends without emphasizing the novelty and contributions of the work in detail.

3. Discussion on Fig. 1 needs significant improvement. For instance, what is PFD, CP1, CP2? What is the expected output from O1 and O2? Divder should be Divider. Better to use f instead of F when referring to frequency, as in f_REF instead of F_REF. AFFCC should be AFLCC.

4. Maintain consistency in the names of the voltages/currents used in the text and the figures. For example, check Fig. 3 where it is Ilock versus I_{lock} in text.

5. Table 1 should have Meas/Sim instead of Meas.Sim. Discussions on Table 1 are completely missing.

6. Mention the limitations of the work in the Conclusions.

Author Response

(The authors gave the same response as above.)

Round 2

Reviewer 2 Report

Thanks to the authors for addressing my comments. I have no further suggestions, except one minor one: A space should exist between "Figure" and its number. For e.g., Fig. 1 instead of Fig.1. Otherwise, the paper can be accepted.